# Assessment of sewer connectivity in the United States and its implications for equity in wastewater-based epidemiology

QinQin Yu[1], Scott W. Olesen[2], Claire Duvallet[2], Yonatan H. Grad[1]*

**1** Department of Immunology and Infectious Diseases, Harvard T. H. Chan School of Public Health, Boston, Massachusetts, United States of America, **2** Biobot Analytics, Inc., Cambridge, Massachusetts, United States of America

* ygrad@hsph.harvard.edu

**Data Availability Statement:** All code and data to reproduce the analyses and simulations are available at https://github.com/gradlab/wastewater_equity.

## Abstract

Wastewater-based epidemiology is a promising public health tool that can yield a more representative view of the population than case reporting. However, only about 80% of the U.S. population is connected to public sewers, and the characteristics of populations missed by wastewater-based epidemiology are unclear. To address this gap, we used publicly available datasets to assess sewer connectivity in the U.S. by location, demographic groups, and economic groups. Data from the U.S. Census' American Housing Survey revealed that sewer connectivity was lower than average when the head of household was American Indian and Alaskan Native, White, non-Hispanic, older, and for larger households and those with higher income, but smaller geographic scales revealed local variations from this national connectivity pattern. For example, data from the U.S. Environmental Protection Agency showed that sewer connectivity was positively correlated with income in Minnesota, Florida, and California. Data from the U.S. Census' American Community Survey and Environmental Protection Agency also revealed geographic areas with low sewer connectivity, such as Alaska, the Navajo Nation, Minnesota, Michigan, and Florida. However, with the exception of the U.S. Census data, there were inconsistencies across datasets. Using mathematical modeling to assess the impact of wastewater sampling inequities on inferences about epidemic trajectory at a local scale, we found that in some situations, even weak connections between communities may allow wastewater monitoring in one community to serve as a reliable proxy for an interacting community with no wastewater monitoring, when cases are widespread. A systematic, rigorous assessment of sewer connectivity will be important for ensuring an equitable and informed implementation of wastewater-based epidemiology as a public health monitoring system.

## Introduction

Wastewater-based epidemiology (WBE) plays an important role in surveillance of SARS-CoV-2 [1, 2], polio virus [3, 4] and other pathogens [5, 6] and has applications to monitoring a

**Funding:** This project has been funded (in part) by contract 200-2016-91779 with the Centers for Disease Control and Prevention (to YG). The findings, conclusions, and views expressed are those of the author(s) and do not necessarily represent the official position of the Centers for Disease Control and Prevention (CDC). The funders had no role in study design, data collection and analysis, decision to publish, or preparation of the manuscript.

**Competing interests:** SWO and CD are former employees of Biobot Analytics, Inc., where CD also holds stocks and bonds. YHG is a board member of Day Zero Diagnostics and an advisor/consultant to GSK, plc.

variety of other public health concerns [7], including opioid usage [8]. One proposed benefit of wastewater-based epidemiology is that wastewater data is more representative of the population than case reporting, which can be biased towards those with health-seeking behavior or access to healthcare [9]. For example, the populations served by the Health and Human Services SARS-CoV-2 National Wastewater Surveillance System were more representative of the entire US's age distribution and Black and Hispanic populations than the vaccinated population [10].

While WBE offers convenient sampling of populations served by public sewers, about 20% of individuals in the US live in homes not connected to public sewers [11, 12]. This includes those on decentralized wastewater systems and those with no wastewater treatment systems. The most common decentralized wastewater system is septic tanks [13], which collect and treat wastewater onsite, typically the yard of the home. Households without wastewater treatment systems may have outhouses or privies, chemical toilets, or no plumbing. Variability in sewer connectivity exists across the U.S. [14, 15], with more connectivity in urban areas than rural areas [11, 16–18] and disparities driven by structural inequalities [19].

As with any emerging public health tools, it is important to ask to what extent the tool exacerbates or alleviates inequities. A few studies have evaluated equity of sewer connectivity on broad geographic scales or at the sewershed level. A 2017 study from Environmental Protection Agency Office of Water showed that households in the U.S. that earned less than the national median household income (MHI, $61,000) were almost 10% more likely to have a decentralized wastewater system or no wastewater treatment system compared to households that earned more than the MHI [20]. Additionally, this study found that as household income decreased, decentralized wastewater system usage increased in Florida, Hawaii, and Delaware, but not in Rhode Island. Based on the 2019 U.S. Census Bureau's American Housing Survey, rural areas are less connected to sewers, and while the income in rural areas was lower than that in metropolitan areas, households connected to septic tanks were wealthier than those connected to sewers within both rural and metropolitan areas [11]. In sewersheds in North Carolina with comprehensive sewershed maps, the sewered population had higher social vulnerability, more minorities, lower income, and lower educational attainment than the unsewered population [21].

However, we have an incomplete understanding of the factors associated with sewer connectivity across the US and the implications for the interpretation of wastewater data. In this study, we sought to address the following questions, focusing our analyses on the U.S.: (1) To what extent is there demographic and economic inequity in sewer connectivity? (2) Which geographic areas have low sewer connectivity? (3) What is the applicability of WBE data to neighboring unsampled communities? To address the first question, we analyzed household-level data of sewer connectivity stratified by geographic, demographic, and economic variables. For the second question, we evaluated datasets aggregated at the county or county subdivision levels to qualitatively identify geographic areas that have low sewer connectivity. To address the third question, we used a mathematical model to simulate WBE in two interacting populations.

## Methods

### Dataset compilation

We assembled publicly available datasets on sewer connectivity, septic connectivity, or no plumbing at the household level across the U.S. from federal and state agencies informed by (1) discussions with experts from federal and state agencies and (2) web searches using terms including "Number of people connected with treatment plants in United States", "Number of

households on septic", and "U.S. census sewer". The types of data included household-level data, locations of sewer systems and population served, and population served by or lacking wastewater systems aggregated by county or county subdivision. All datasets are summarized in Table 1 and described in more detail in the Supplementary Information.

### Calculation of weights, error bars, and summary statistics in American Housing Survey (AHS) data

All summary statistics were calculated by applying the weights reported in the AHS Public Use File [26]. Because not every household was sampled and there was uneven sampling of households across the U.S., the weight estimates the number of similar households that each surveyed household represented. Error bars on summary statistics represent the middle 95% of values calculated using the reported replicate weights (160 replicate weights per household). Each summary statistic was calculated for each replicate separately before taking the middle 95% of values. Only categories with at least 5 households were used.

### Analysis of sewer connectivity by urban or rural areas in AHS data

We used the most recent AHS National Sample with publicly available data on the urban or rural location of the household (2013) which categorized households into being in a central city of a Metropolitan Statistical Area (MSA), in an urban area of an MSA but not in a central city, in a rural area of an MSA but not in a central city, in an urban area outside of an MSA, or in a rural area outside of an MSA [34]. Central cities are defined as having either (1) $\geq$250,000 population or at least 100,000 people working within corporate limits, (2) $\geq$25,000 population, at least 75 jobs for each 100 residents who were employed, and 60% or fewer of the city's resident workers commuted to jobs outside, or (3) 15,000–25,000 population, at least one third the size of the metropolitan statistical area's largest city, and met the two commuting requirements in (2). Metropolitan Statistical Areas were defined as whole counties that have significant levels of community and contiguous urban areas in common and may include rural areas. Urban areas were defined as having $\geq$2500 people, with at least 1500 residing outside of institutional group quarters. Rural areas were defined as those not in urban areas.

### Analyses of geographic areas with low connectivity to sewer

Data of individual locations of septic tanks or sewer collection systems, when they existed, were aggregated based on latitude and longitude into county or county subdivision boundaries as defined by the U.S. Census' TIGER shapefiles of the corresponding year [35]. To calculate the fraction of a geographic area connected to septic or sewer, the total population size or total number of households in that geographic area was taken from the corresponding year in the American Community Survey (ACS) 5-year estimate [27]. For county subdivision and census designated place analyses, only areas with at least 5 households and 20 population size were used for the analyses. This thresholding removed at most 1% of the total population and at most 0.03% of the total number of households, making it unlikely to bias our results. Detailed data analyses of the state and island area datasets are described in the Supplementary information.

Counties and county subdivisions were classified into Metropolitan statistical areas (at least one urbanized area of 50,000 or more population), Micropolitan statistical areas (at least one urban cluster of 10,000–50,000 population), or Rural areas (all other areas) using the U.S. Census classification of the encompassing Core Based Statistical Area (core area containing a substantial population nucleus together with adjacent territory that has a high degree of social and economic integration with the core as measured by commuting ties) [36].

**Table 1. Datasets used in this study.**

| Dataset | Location | Year(s) | Description | Sampling method | Dataset completeness and potential biases | Ref. |
|---|---|---|---|---|---|---|
| American Housing Survey | US excluding Island areas | 2013, 2019, 2021 | This was a longitudinal survey of housing, demographic, and economic characteristics of approximately 60,000–80,000 representative households. | The sampling method was a representative sample of all U.S. households at the time of sample selection (most recently in 2015 and before that, in 1985). New housing units were added each survey cycle. | The survey excludes group quarters, businesses, hotels, and motels. Additionally, the survey was more likely to classify rural seasonal homes as vacant units, counted large cluster septic systems as public sewer, and was thought to lose new households built in rural areas due to its longitudinal design. | [12, 22–26] |
| American Community Survey 5-year estimates | US excluding Island areas | 2021 | This was a monthly survey of characteristics of populations and households of a representative subsample of addresses in the US. The survey reports the number of households lacking complete plumbing facilities by geographic subdivision. | The survey was sent to a random sample of addresses in the U.S. every month. Each address was selected no more than once every 5 years. | The surveys were completed by self-response (although a subset of non-responses were followed up with personal visits). Responses were reviewed for completeness and questionnaires needing clarification were followed up by phone calls. The data was averaged over 5 years to reduce statistical fluctuations. | [27] |
| Florida Department of Health septic tank inspections | Florida | Up to 2012 | This dataset includes the onsite sewage treatment and disposal systems locations inspected by the Florida Department of Health reported in June 2012. | The sample included onsite sewage treatment and disposal system permits recorded in the Florida Department of Health system. | As of 2022, a permit was required for construction and inspection was recommended every 3–5 years after. From this dataset, an estimated 7% of households were connected to septic tanks across the state. This was substantially less than the Florida Department of Environmental Protection's estimate that approximately one third of Florida's population used septic tanks [28]. | [29] |
| EPA Clean Watersheds Needs Survey | US | 2012 | This survey included voluntary submissions of the locations of publicly owned wastewater collection and treatment facilities and their estimated population served for the purposes to assess funding needs for treatment works projects. | The responses were coordinated by states. | This survey had voluntary responses and variable effort and resources that each state put into the survey. States that had the most comprehensive responses were New York, California, Florida, New Jersey, Maryland, Iowa, Minnesota and Michigan. Missing facilities included those in South Carolina, the Northern Mariana Islands, and American Samoa; facilities whose projects did not have documented solutions or cost estimates; privately owned wastewater facilities; facilities on tribal lands and Alaskan Native Villages; and facilities whose projects received funding from other sources. Small community facilities are thought to be underrepresented due to having less resources for completing the survey, but this was not quantified. | [30, 31] |
| US Census Island Areas Decennial Survey | US Island Areas | 2020 | This was a survey of housing, social, and economic information of all housing units. | The survey was conducted in person and via phone interviews. | The COVID-19 pandemic impacted collection of data on group quarters [32] and the survey had higher nonresponse rates than previous years' surveys, particularly for Guam. Quality control checks were performed on responses. | [33] |

The datasets are further described in the S1 Appendix and the dataset completeness and potential biases are further assessed in the S2 Appendix. Additional datasets that are briefly described in the main text but primarily analyzed in the Supplementary information are summarized in S1 Table.

### Demographic and economic variable correlations with sewer or septic connectivity

To assess correlations of demographic and economic variables with sewer and septic connectivity when aggregating at the county, county subdivision, or census designated place levels, we used the ACS 5-year estimates from the corresponding year and geographic scale. Only geographic subdivisions with at least 5 households and 20 population size were used. The Pearson correlation coefficient between demographic or economic variables of interest and the percentage of households in a geographic subdivision connected to septic tanks or sewer was calculated using the stats.pearsonr function in the scipy package (version 1.10.1) in Python (version 3.11.0). A Bonferroni correction was applied to correct for multiple hypothesis testing to calculate the q values.

### Simulations of interacting populations and wastewater sampling

Two interacting populations (A and B) were modeled in a deterministic compartmental model with susceptible, infected, and recovered (SIR) compartments for each population and a fraction, $\varepsilon$, of cross-population contacts. The base parameters were set at population size $N_A = N_B$ = 5000, recovery rate $\gamma_I$ = 0.18 inverse days (corresponding to an infectious period of 5.6 days), basic reproduction number $R_0^A = R_0^B = \frac{\beta_A}{\gamma_I} = \frac{\beta_B}{\gamma_I} = 1.5$ and were varied in sensitivity analyses. The differential equations were solved using the integrate.odeint function in the scipy package (version 1.6.2) in Python (version 3.11.0) using a timestep of 0.1 days.

Wastewater sampling was modeled by the number of copies of pathogen genetic material shed per day by each population that was sampled by wastewater divided by the volume of sampled wastewater produced per day by each population. Both numerator and denominator depended on the fraction of waste produced (including shed pathogen genetic material) by each population that is sampled by wastewater, $f_A$ and $f_B$. A detailed description of the simulations is provided in S1 Appendix.

### Code availability

All code and data to reproduce the analyses and simulations are available at https://github.com/gradlab/wastewater_equity.

## Results

### Is there inequity in who could be sampled by wastewater-based epidemiology?

To address this question, we used the American Housing Survey (AHS), a biannual longitudinal survey of representative housing units within the U.S that reports household-level data of sewer connectivity, demographic factors, and economic factors (Table 1). We focused our analyses on geographic scales with a reliable sample size for statistical analyses (census division, urban/rural at the national level, metropolitan statistical areas) [37].

According to the 2021 AHS, 83% of US households were connected to public sewers and 16% to standard septic tanks (S1 Fig). Each of the other forms of sewage disposal (non-standard septic tanks, chemical toilets, outhouses or privies, other, none, or not reported) was more than an order of magnitude less common (S1 Fig). Across census divisions in 2021, the western US was overall better connected to sewers than the eastern US, with the highest levels of sewer connection in the Pacific (91.3% of households, CI: 90.8%-91.7%) and Mountain (87.6% of households, CI: 86.8%-88.4%) divisions and the lowest levels of sewer connection in

the New England (72.9% of households, CI:72.1%-74.0%) and East South Central (74.9% of households, CI: 73.7%-76.3%) divisions (Fig 1).

Across census divisions, households with an Asian, Black or African American, or Native Hawaiian and Other Pacific Islander householder (the owner or renter of the unit) were on average 14.0%, 13.1%, and 12.5% more connected to sewers, respectively, than the overall census division; however, specifically in the Pacific census division, households with a Native Hawaiian and Other Pacific Islander householder were 5.3% less connected to sewers than the overall census division (Fig 2). Households with an American Indian and Alaska Native or White householder were both on average 2.6% less connected to sewers than the overall census division. Across census divisions, households with a Hispanic householder were on average 11.7% more connected to sewers whereas households with a non-Hispanic householder were 1.1% less connected to sewers than the overall census division.

Connection to sewers decreased with the age of the householder (Fig 2), with the exception that households with a householder aged 75+ were better connected in most census divisions than those with a householder aged 65–74 (79.7% vs 77.9% of households). Connection to

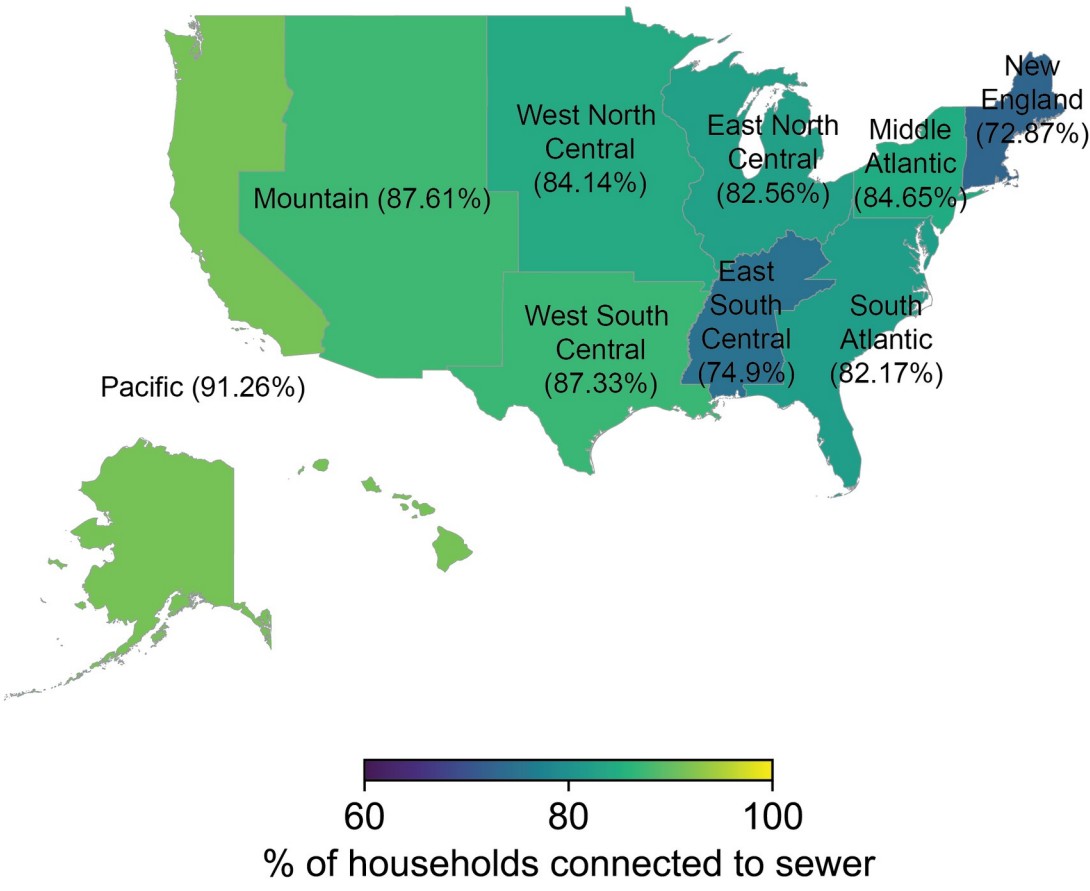

**Fig 1. The percentage of households connected to sewers in each census division.** Data are from the 2021 American Housing Survey National Survey [12]. The number of sampled households for each census division are 3890 (New England), 7018 (Middle Atlantic), 8703 (East North Central), 2862 (West North Central), 13643 (South Atlantic), 2903 (East South Central), 8051 (West South Central), 4570 (Mountain), and 12501 (Pacific). The map base layer is taken from the U.S. Census 2021 Cartographic Boundary File by U.S. Divisions (https://www2.census.gov/geo/tiger/GENZ2021/shp/cb_2021_us_division_500k.zip). The cartographic boundary files are simplified representations of the U.S. Census TIGER shapefiles (terms of use: https://www2.census.gov/geo/pdfs/maps-data/data/tiger/tgrshp2021/TGRSHP2021_TechDoc_Ch1.pdf).

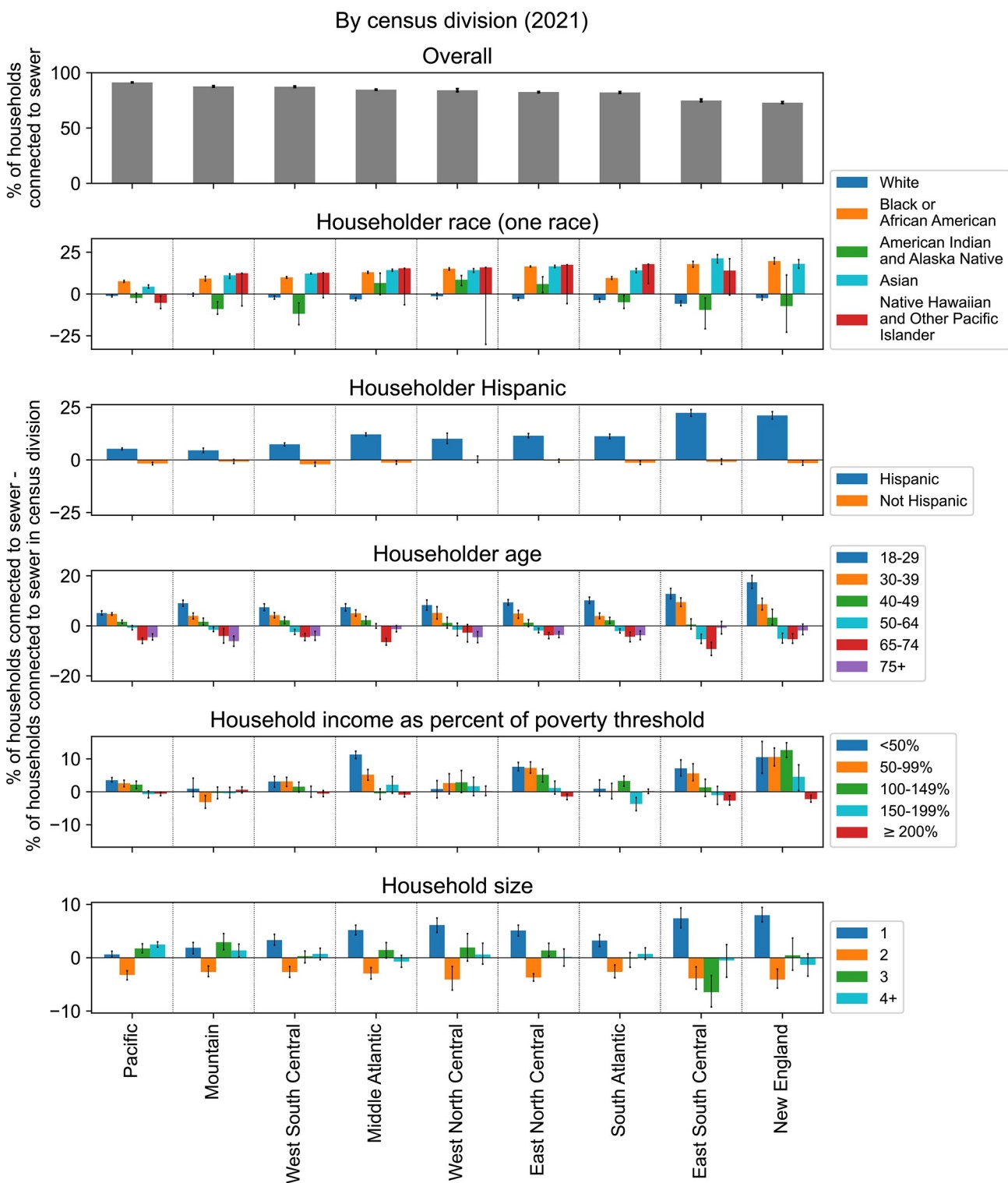

**Fig 2. The percentage of households connected to sewers in each census division by demographic and economic characteristics.** Data are from the 2021 American Housing Survey National Survey [12]. The number of sampled households for each category is shown in S2 Fig.

sewers decreased with household income, as measured as a percentage of the federal poverty threshold determined by the Census [38], with the exception of the Mountain region where households with an income at 50–99% of the poverty level were less connected than those with income at ≥200% of the poverty level (84.5% vs 88.2% of households). Additionally, households with 1 person were on average 4.2% more connected whereas households with 2 people were on average 3.6% less connected to sewers than the overall census division. There were no consistent trends across census divisions for households with more than 2 people.

We sought to describe how the differences in sewer connection between demographic and economic groups observed at the census division level was affected by household location in urban or rural areas (see Methods). The qualitative trends observed at the census division level for race, ethnicity, age, household size, and income were largely preserved across national urban and rural areas (Fig 3). One notable exception was that lower sewer connectivity for households with an American Indian and Alaska Native or Hawaiian and Other Pacific Islander householder compared to those with a Black or Asian householder was only observed in locations inside of an Metropolitan Statistical Area (MSA) but outside of a central city (both urban and rural). In central cities and locations outside of an MSA (both urban and rural), households with an American Indian and Alaska Native or Hawaiian and Other Pacific Islander householder had similar levels of sewer connectivity as households with a Black or Asian householder.

Next, we assessed sewer connectivity in large metropolitan areas, which was the smallest geographic scale resolved by the AHS due to sample size. In 35 of the largest metropolitan areas in 2019 and 2021, household connectivity to sewers ranges from 69.2% (Birmingham-Hoover, AL) to 99.4% (LA-Long Beach-Anaheim, CA) (S4 Fig). In each metropolitan area, households with a White or American Indian and Alaska Native householder, with an older householder, with higher income, with a larger household size, and with a non-Hispanic householder were less connected to sewers than the average household, which was similar to what was seen when stratifying by census division. Exceptions were lower connectivity for households with a Hispanic householder in San Jose, CA (Hispanic householder: 96.1% of households, CI: 95.1%-97.2%; non-Hispanic householder: 97.8% of households, CI: 97.5%-98.2%) and lower connectivity for households with lower income in San Francisco, CA; Dallas, TX; Chicago, IL; and Tampa, FL (S4 Fig).

In summary, at the census division level, broad trends of sewer connectivity by racial group revealed higher connectivity for households with an Asian, Black or African American, Native Hawaiian and Other Pacific Islander, or Hispanic householder and lower connectivity for households with an American Indian and Alaska Native or White householder. Sewer connectivity decreased with age, decreased with household income, and was highest in households with one person. These trends were similar when stratifying by urban, rural, central city, and MSA status, except that the broad racial disparities described above were only observed in urban and rural areas that are inside of an MSA but outside of a central city. Broad trends were also similar for individual large metropolitan areas, with some exceptions for households with a Hispanic householder and trends by household income in particular MSAs.

## Which geographic areas have low connectivity to sewers and what are their demographics?

Next, we sought to assess which geographic areas in the U.S. have low connectivity to sewer systems. These are areas where wastewater-based epidemiology may provide little direct information on disease dynamics. Additionally, we assessed correlations of sewer connectivity at the county or county subdivision level with demographic and economic factors. In the absence

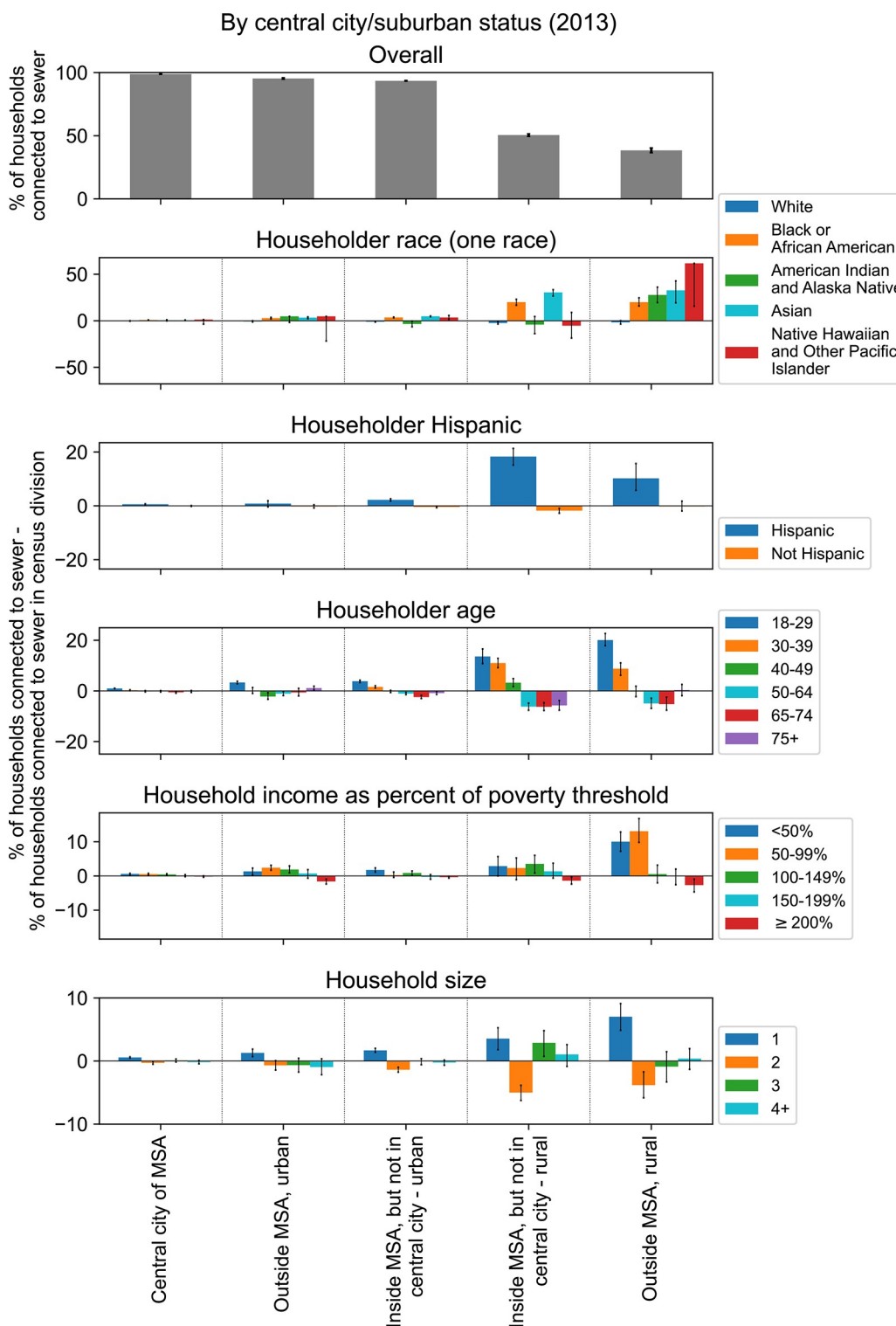

**Fig 3. The percentage of households connected to sewers by urban and rural categorizations and by demographic and economic characteristics.** See Methods for definitions of Metropolitan Statistical Area (MSA), central city, urban, and rural. Data from the 2013 American Housing Survey National Survey [39]. The number of sampled households for each category is shown in S3 Fig. MSA: metropolitan statistical area.

of comprehensive national data on sewer connectivity, we assembled publicly available state-based datasets, which included California, Florida, Iowa, Maryland, Michigan, Minnesota, New Jersey, and New York, as well as the U.S. Island Areas. The compilation of the datasets and an assessment of data completeness are described in the S2 Appendix, Table 1, and S1 Table.

Using data from the 2021 American Community Survey on the percentage of occupied housing units lacking complete plumbing facilities by county, we defined the lower bound on the percentage of the population not connected to sewers. In 14 counties, ≥10% of occupied housing units lacked complete plumbing facilities; of these counties, 12 were in Alaska and 2 were in the Navajo Nation. In 5 counties, ≥20% of occupied housing units lacked complete plumbing facilities, all in Alaska. The Yukon-Koyukuk Census Area in Alaska, had the highest percentage of occupied housing units lacking complete plumbing facilities at 36% (S5 Fig, S2 Table).

The Florida Department of Health onsite sewage treatment and disposal systems inspection data (Fig 4) revealed that the percentage of households not on septic tanks (expected to

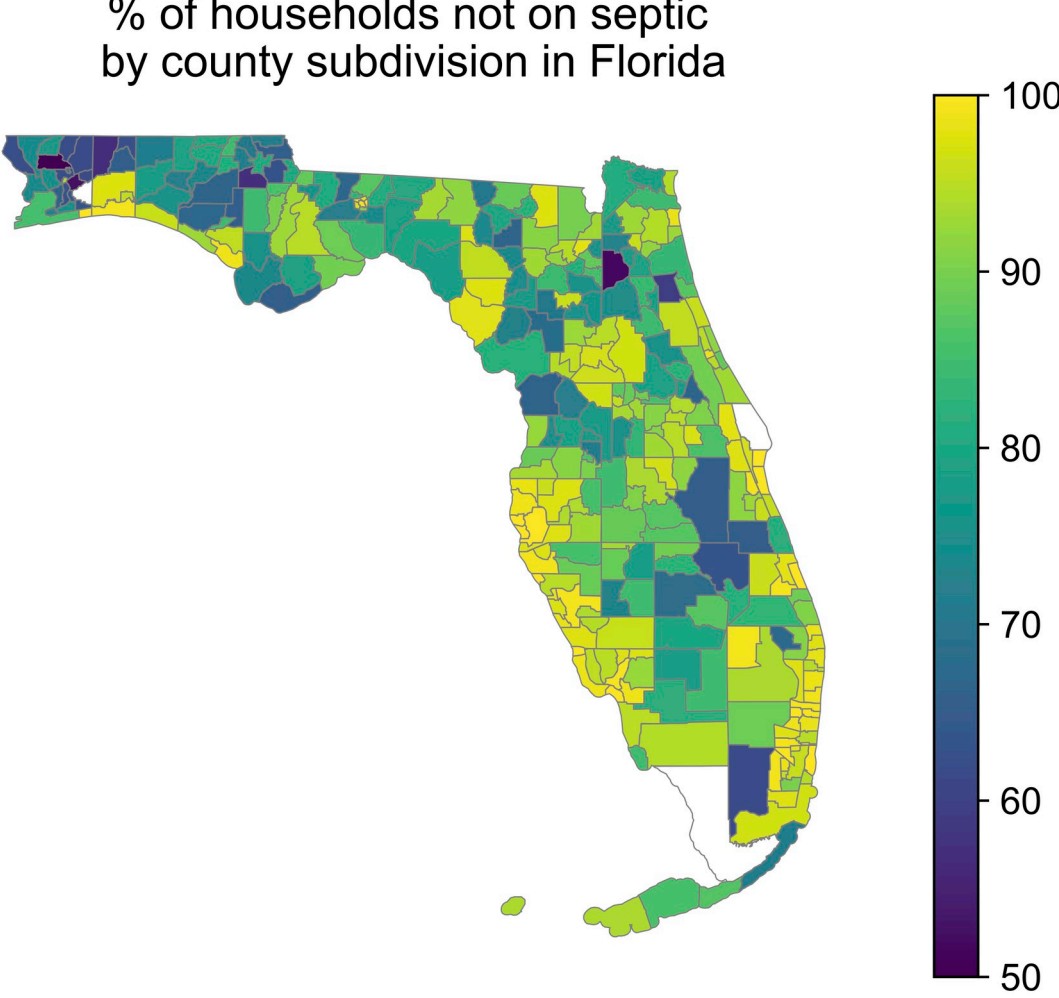

**Fig 4. Map of the percentage of households in each county subdivision of Florida *not* connected to septic tanks.** Data are from the Florida Department of Health septic tank inspection data from June 2012. County subdivisions with fewer than 5 households or 20 population size (<0.01% of total households and <0.01% of total population size in the state) are displayed in white. The map base layer is taken from the U.S. Census 2012 TIGER/Line Shapefile by Florida County Subdivision (https://www2.census.gov/geo/tiger/TIGER2012/COUSUB/tl_2012_12_cousub.zip; terms of use: https://www2.census.gov/geo/pdfs/maps-data/data/tiger/tgrshp2012/TGRSHP2012_TechDoc_Ch1.pdf).

correlate with the percentage of households connected to sewer) was higher in metropolitan compared to micropolitan county subdivisions and higher in micropolitan compared to rural county subdivisions, but the effect was not significant (p = 0.052 and p = 0.37, respectively) (S6a and S6b Fig). The percentage of households not on septic tanks was significantly lower in county subdivisions in the panhandle (74.0% vs 92.0%, $p<10^{-9}$) and in county subdivisions that did not border the coast (86.6% vs 95.7%, $p<10^{-9}$) (S6 Fig). By combining this dataset with the 2012 American Community Survey (ACS) describing demographic and economic characteristics of county subdivisions, we found that county subdivisions with less septic usage (suggesting more sewer connectivity) were significantly more Hispanic, more Asian, and had a smaller household size (Table 2 and S7 Fig), consistent with the nation-wide results from the AHS. These trends were driven by metropolitan county subdivisions (S8 Fig), and an additional trend in metropolitan county subdivisions not observed for all county subdivisions was that county subdivisions with less septic usage had a significantly lower percentage of American Indian and Alaska Natives. There were too few micropolitan and rural county subdivisions to observe significant trends (S9 and S10 Figs).

In the 8 states with robust data in the EPA Clean Watersheds Needs Survey, multiple neighboring counties in Florida, Michigan, and Minnesota had less than 20% of residents receiving sewage collection, whereas this was not the case in California, Florida, Iowa, Maryland, New Jersey, and New York (S11 Fig). Across multiple states, sewer connectivity by county was positively correlated with percent Asian and percent Black or African American and negatively correlated with percent White (Table 3, full table shown in S3 Table), consistent with the nationwide trends from the AHS data. While counties more connected to sewers appeared to have a significantly lower percent of American Indian and Alaskan Natives in California, this is affected by missing data from Indian reservations. Age and percent without health insurance were negatively correlated with sewer connectivity. Counties more connected to sewers had significantly higher income in Minnesota, Florida, and California, in contrast to the national

**Table 2. Correlation of the percentage of a Florida county subdivision *not* connected to septic tanks with different demographic or economic variables.**

| Demographic or economic variable | Correlation of variable with % of county subdivision *not* on septic (Pearson correlation coefficient) | Significance (q-value) |
|---|---|---|
| Percent one race and Asian | 0.31 | **<0.001** |
| Percent Hispanic | 0.21 | **<0.01** |
| Average household size | -0.19 | **<0.01** |
| Percent one race and American Indian and Alaska native | -0.15 | 0.07 |
| Median age | 0.12 | 0.36 |
| Percent one race and White | -0.12 | 0.40 |
| Percent one race and Black or African American | 0.09 | 1.11 |
| Median income | 0.07 | 2.27 |
| Percent one race and some other race | 0.07 | 2.33 |
| Percent one race and Native Hawaiian and other Pacific Islander | -0.04 | 5.76 |
| Percent uninsured (health insurance) | -0.02 | 8.24 |

Correlations significant to 5% are shown in bold. Data are from the Florida Department of Health septic tank inspection permits reported in 2012.

**Table 3. Correlation of the percentage of households in a county connected to public sewers with different demographic or economic factors by state.**

| Demographic or economic variable | State | Correlation of variable with % of households by county connected to public sewers (Pearson correlation coefficient) | Significance (q-value) |
|---|---|---|---|
| Median age | MN | -0.43 | <0.01 |
| | MI | -0.41 | 0.01 |
| Median income | MN | 0.44 | <0.01 |
| | FL | 0.48 | <0.01 |
| | CA | 0.49 | <0.01 |
| Percent one race and American Indian and Alaska Native | CA | -0.50 | <0.01 |
| Percent one race and Asian | MN | 0.58 | <0.001 |
| | MI | 0.51 | <0.001 |
| | NY | 0.55 | <0.001 |
| | CA | 0.56 | <0.001 |
| | FL | 0.53 | <0.001 |
| Percent one race and Black or African American | MN | 0.62 | <0.001 |
| | MI | 0.59 | <0.001 |
| | NY | 0.55 | <0.001 |
| | IA | 0.38 | <0.01 |
| | CA | 0.46 | 0.02 |
| Percent one race and White | MI | -0.57 | <0.001 |
| | CA | -0.54 | <0.001 |
| | NY | -0.51 | <0.01 |
| | IA | -0.35 | 0.03 |
| Percent uninsured (health insurance) | MI | -0.38 | 0.03 |

Only correlations that are significant (q-value $\leq$ 0.05) are shown. Data are from the 2012 EPA Clean Watersheds Needs Survey.

results from the AHS data. Significant trends of sewer connectivity with demographic and economic factors were predominantly driven by metropolitan counties (S4 Table).

Analysis of the Island Areas of Guam, the Northern Mariana Islands, the Virgin Islands, and American Samoa revealed lower levels of sewer connectivity than in the states, with considerable spatial variability in connectivity (see S2 Appendix).

## Factors influencing applicability of wastewater-based epidemiology to communities lacking sewer connection

The inequities in sewer connections among communities raise the question of the applicability of inferences drawn from wastewater data in one community to communities lacking sewer connections. To explore this question, we used a deterministic compartmental model of two interacting populations with tunable levels of interaction, each with susceptible-infected-recovered (SIR) dynamics, and with wastewater sampling in each population into a common sample (Fig 5a). We set population A to be entirely connected to sewers and population B to have a tunable level of sampling by wastewater.

A common application of wastewater data is to aid in determining when an outbreak has peaked, which can inform policy decisions on when to ease restrictions. We first asked how well the wastewater data could predict the time of peak infections in population B when it is completely unconnected to sewer systems and thus the only sampling is from population A. We found that when the two populations have similar sizes, the wastewater concentration

peak and the infection peak in population B was within one generation time except when the interactions between the two population is weak (Fig 5b). The generation time is the average time between an individual's infection and transmission, which in an SIR model is the same as the infectious period. For example, in two populations with 5000 individuals each, a disease basic reproduction number of 1.5, and infectious period of 5.6 days, the percentage of contacts that occur across populations must drop below 4% before the peak in wastewater concentration occurred more than one generation time apart from the peak in infections in the unconnected population (Fig 5c).

We then weakened the assumption that population B had no contribution to the sampled wastewater and allowed a fraction of population B to be sampled by wastewater. For example, individuals in population B may commute to a workplace sampled by wastewater, or a fraction of population B may be connected at home. As the fraction of population B that is sampled by wastewater increases, the time between infection and wastewater peaks decreases; however, the fraction of B that is sampled has a much weaker effect on the wastewater peak time than the interaction parameter between the communities. For example, with the above parameters

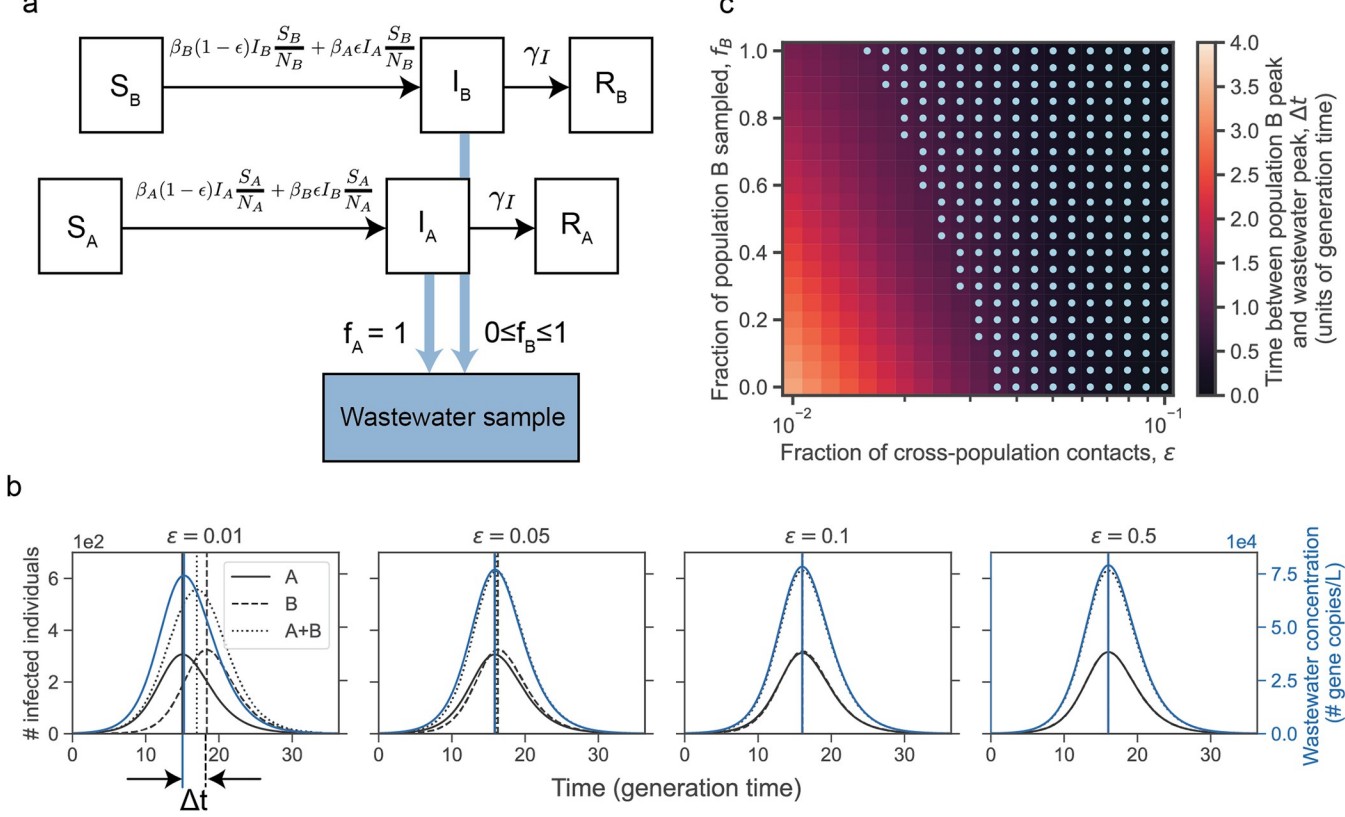

**Fig 5. Deterministic compartmental model of two interacting populations with susceptible, infected, and recovered compartments and sampling by wastewater.** (a) Schematic of model. Population A is entirely sampled by wastewater and a fraction of population B, $f_B$, is sampled by wastewater. (b) Number of infected individuals (black curves) in population A, population B, and population A and B combined and wastewater concentration (blue curves) over time for different interaction strengths or fractions of cross-population contacts ($\varepsilon$). The vertical lines indicate the peak time for the curves with matching line style and color. The time between the population B peak in infections and the wastewater concentration peak is indicated by $\Delta t$. (c) The time between the population B peak in infections and the wastewater concentration peak (defined in (b)) when varying the fraction of cross-population contacts ($\varepsilon$) and the fraction of population B that is sampled ($f_B$). Light blue dots indicate parameter regimes in which $\Delta t$ is less than 1 generation time. In both (b) and (c), the simulation starts with a single infected individual in population A at time 0 and no infected individuals in population B. The parameters are set to population size $N_A = N_B = 5000$, basic reproduction number $R_0^A = R_0^B = R_0 = \frac{\beta_A}{\gamma_I} = \frac{\beta_B}{\gamma_I} = 1.5$, rate of recovery $\gamma_I = 0.18$, and the overall contact rate times the probability of infection given contact $\beta_A = \beta_A = \gamma_I R_0$.

and equal population sizes, with 10% sampling of population B, the cross-population contacts must drop below 4%, and with 50% below 3% and with 100% below 2% before the wastewater concentration peak and the peak in the number of infections in population B drops below one generation time.

Varying the population size of both populations (with an equal population size in the two populations), the recovery rate or generation time, and the $R_0$ of both populations (with an equal $R_0$ in the two populations) has only a weak effect on the discrepancy between wastewater concentration and population B infection peaks (S12 Fig).

When the unconnected population has a smaller size than the connected population, then even substantial interactions between the two populations can lead the wastewater concentration to peak before the infections in the unconnected population peaks (S12 Fig). Both very weak and very strong interactions cause the wastewater peak to coincide with the peak number of infections in the unconnected population.

Additionally, when the unconnected population has a smaller $R_0$ than the connected population (which could occur for instance if the unconnected population was more spread out geographically and thus had a lower contact rate), the outbreak in population B peaks later than the wastewater peak leading to a discrepancy in the peak times (S12 Fig).

## Discussion

### Equity of sewer connectivity

Our analysis of available datasets revealed considerable variability in sewer connectivity within and between communities, across locations, demographics, and economic statuses in the United States. The western US had higher levels of sewer connectivity than the eastern US. Across the US, lower connectivity to sewers was observed for American Indian and Alaska Native householders; White householders; non-Hispanic householders; older populations; and larger households in most census divisions. In the Pacific census division, lower connectivity to sewers was observed for Native Hawaiian and Other Pacific Islander householders specifically in areas inside of a metropolitan statistical area but outside of a central city. Across census divisions (with the exception of the Mountain census division), households with higher income were less connected to sewers, consistent with nation-wide observations [11]. The decrease in sewer connectivity by income could not be explained by collinearity between age and income as age and income exhibited a weak negative correlation (S13 Fig). These results are also consistent with the observation that in North Carolina, minorities and those with social vulnerabilities are overrepresented in the sewered population [21]. Aggregating by large metropolitan areas yielded similar trends as when aggregating by census divisions, except for lower connection for Hispanic householders in San Jose area and for households with lower income in San Francisco, Dallas, Chicago, and Tampa metropolitan areas, whereas these groups were better connected to sewers at the census division level. The differences in the MSA-level data suggested heterogeneity in connectivity across locations.

We have not attempted to determine why the observed differences in sewer connectivity exist. Many of the variables that we analyzed are likely to be correlated with one another.

### Geographic variation in sewer connectivity

Large parts of Alaska and the Navajo Nation lacked plumbing. This is consistent with the finding from the Annual Report to Congress on Sanitation Deficiency Levels for Indian Homes and Communities in 2019 that approximately 20% of Indian homes in Alaska do not have sewage disposal and about 10% of Indian homes in the Navajo Nation do not have sewage disposal [40].

State-level datasets revealed that sewer connectivity varied between and within communities. While in some states most areas had sewer connectivity (California, Iowa, Maryland, New Jersey, New York), in others there were large regions that lacked connectivity (Minnesota, Michigan, Florida). Asian and Hispanic populations were less connected to septic tanks across county subdivisions in Florida; however, there was considerable variability across the state and potential biases in the data. In contrast to the national data, in Minnesota, Florida, and California, counties with lower median household incomes were less connected to sewers, suggesting variability across the nation. This result in Florida is consistent with the observation that as income levels rose, household decentralized system usage declined in Florida [20]. However, we did not observe any significant correlation of income with septic usage in the Florida Department of Health onsite sewage treatment and disposal systems inspection permitting dataset, possibly due to not all septic tanks having a permit that is reported in this dataset.

While these correlations may be largely driven by differences between urban and rural areas within a county or county subdivision, they may still be very useful in assessing inequities in sewer connectivity in large catchments that span neighboring counties or county subdivisions.

## Data limitations

Substantial data gaps and biases prevented a comprehensive analysis of disparities in sewer system connectivity within and between communities across the US, particularly for small communities, tribal lands and Alaska Native Villages, and state and local geographic scales. The current design of the American Housing Survey has too small of sample size to study state and local geographic scales [37]. Additionally, the AHS may have underestimated the fraction of households using septic systems due to its survey design (Table 1). The differences between Florida state data supports the need for detailed and reproducible longitudinal studies of sewer connectivity (both in developing sampling and analysis methods). Additionally, multiple datasets were over a decade old, and changes to which households are served by sewer and septic systems may have occurred since the datasets were collected.

Public health officials may find it beneficial to conduct local equity studies of sewer connectivity. Data on the catchment area sizes of the wastewater treatment plants would allow a better understanding of the geographic extent of the sampled population and the level of geographic aggregation needed to study sewer connectivity inequities. A recent study mapping wastewater treatment plant catchment areas and their population sizes served in New York state provides a template for how to create these sewer catchment maps from a combination of permitting, survey, and tax record data [16]. If possible, it would be helpful to consolidate and standardize existing data where it exists (for instance, sewage disposal permitting at the local or state levels). Additionally, if the EPA's request to expand the question on the ACS about access to plumbing facilities to additionally ask about the type of plumbing facility [20, 41] is granted, then this would be a valuable dataset with the geographic and temporal resolution to study equity in sewer connectivity moving forward.

## Generalizability of wastewater-based epidemiology data in light of heterogeneities in sewer connectivity

Our modeling results suggest that even weak interactions between two communities allow wastewater monitoring in one community to serve as reliable proxy for the time of maximum infections in the other community when the population sizes and $R_0$ of the two populations are comparable, but not when the unconnected population has a substantially lower population size or $R_0$. In the scenario with unequal population size, outbreaks that are seeded at the

same time in the two populations in the absence of interactions peak earlier in the unconnected population due to its smaller size. With weak interactions, seeding of infections from population A to B occurs slower, leading to more coincidence in peaks; with strong interactions, then the dynamics of B are dominated by those in A, leading to more coincidence in peaks; at intermediate interaction strengths, infections are seeded earlier and peaks early, leading to the largest discrepancy in wastewater and infection peaks.

As the purpose of our model was to explore the impact of factors on the generalizability of wastewater rather than to accurately capture the dynamics in all scenarios, we used a compartmental susceptible-infected-recovered model (SIR) and made simplifying assumptions, including that all infected individuals shed the same amount of pathogen genetic material into wastewater and shed only during the infected period; that all individuals contribute equally to wastewater; and that pathogen detection in wastewater has perfect sensitivity. We also note that compartmental models have been used to analyze wastewater data [42].

Our results suggest that in assessing the generalizability of wastewater data, it would be useful to estimate the extent of mobility between connected and unconnected communities. Interestingly, Ref [43] found no correlation between the size of a catchment area and the correlation of wastewater with case data for SARS-CoV-2, consistent with the result that interactions between connected and unconnected communities cause the disease dynamics to look similar to the wastewater data. In regions without sewer connectivity or with little interaction between connected and unconnected communities, wastewater data from neighboring communities will be less informative. In areas with low sewer connectivity in households, sampling wastewater outflow at frequently visited non-household locations (i.e., schools, offices, malls, etc.) may capture a more representative population.

## Additional considerations for equity in WBE

In addition to inequities in sewer connectivity, the sewer locations used for wastewater sampling should be considered to promote demographic equity and ensuring the ability to capture spatiotemporal trends [21, 44–46]. While we have focused on analysis on the US due to data availability, internationally disadvantaged populations are associated with lower access to sewers [47–49]. The ongoing development of wastewater sampling in non-sewered settings, for example in water channels in Las Vegas Valley [50], onsite sanitation facilities in Bangladesh [51], a refugee camp in Lebanon [52], and various non-sewered settings in low and middle income countries [53–55], represents a critical area for research and development.

## Conclusions

In summary, while wastewater-based epidemiology is a useful tool to monitor disease burden and dynamics, our analyses suggest that access to this new tool varies across the US. More comprehensive data on sewer connectivity is needed, and in combination with assessments of mobility and population parameters, these data can help with the design of wastewater sampling schemes and the interpretation for epidemic trends in sampled and neighboring unsampled communities.

## Supporting information

**S1 Appendix. Supplementary methods.**
(DOCX)

**S2 Appendix. Supplementary analyses.** Compilation of data, assessment of data completeness, sewer connectivity in U.S. Island Areas, supplementary figures, and supplementary

tables.
(DOCX)

**S1 Fig. Percentage of households connected to various wastewater processing systems.** The number of households sampled (n) for each category are also shown. Data are from 2021 American Housing Survey.
(PNG)

**S2 Fig. The percentage of households connected to sewers in each census division by demographic and economic characteristics (same as Fig 2) including the number of sampled households in each category.**
(PNG)

**S3 Fig. The percentage of households connected to sewers by urban and rural categorizations and by demographic and economic characteristics (Fig 3) including the number of sampled households in each category.**
(PNG)

**S4 Fig. Top 35 metropolitan areas' connectivity to public sewers by demographics.** Data are from the 2019 and 2021 U.S. Census American Housing Survey [12, 23]. The number of sampled households for each category can be found in the Github repository in outputs/ahs_sewer_connectivity_by_cbsa.csv in the "num_observations" column.
(PNG)

**S5 Fig. Percent of occupied housing units lacking complete plumbing facilities by county.** Data are from the 2021 U.S. Census American Community Survey. The map base layer is taken from the U.S. Census 2021 TIGER/Line Shapefile by U.S. County (https://www2.census.gov/geo/tiger/TIGER2021/COUNTY/tl_2021_us_county.zip; terms of use: https://www2.census.gov/geo/pdfs/maps-data/data/tiger/tgrshp2021/TGRSHP2021_TechDoc_Ch1.pdf).
(PNG)

**S6 Fig. Comparison of percentage of households not on septic across different geographies in Florida.** Data are from the Florida Department of Health septic tank inspection permits reported in 2012. (a) Map of county subdivision colored by whether it is a Metropolitan Statistical Area (blue), Micropolitan Statistical Area (orange), or Rural Area (neither metropolitan not micropolitan, green). (b) Percentage of households not on septic in county subdivisions stratified by whether the county subdivision is a Metropolitan Statistical Area, Micropolitan Statistical Area, or Rural Area. County subdivisions that are Metropolitan Statistical Areas have a higher percentage of households not on sewer than those that are Micropolitan Statistical Areas, but the effect is not significant (p = 0.052). County subdivisions that are Micropolitan Statistical Areas have a higher percentage of households not on sewer than those that are Rural Areas, but the effect is not significant (p = 0.37). (c) Map of county subdivisions colored by whether they are always included in references to the Florida panhandle. Orange: county subdivisions in the panhandle. Blue: county subdivisions not in the panhandle. (d) Percentage of households not on septic in county subdivisions stratified by whether the county subdivision is in the panhandle. Each point represents a county subdivision. County subdivisions in the panhandle have a significantly lower percentage of households not on septic ($p < 10^{-9}$). (e) Map of county subdivisions colored by whether they have a coastline. Orange: county subdivision with coastline. Blue: county subdivisions without coastline. (f) Percentage of households not on septic in county subdivisions stratified by coastal and not coastal county subdivisions. Each point represents a county subdivision. County subdivisions not on the coast have a significantly higher fraction of households on septic ($p < 10^{-9}$). The map base layers in (a), (c),

and (e) are taken from the U.S. Census 2012 TIGER/Line Shapefile by Florida County Subdivision (https://www2.census.gov/geo/tiger/TIGER2012/COUSUB/tl_2012_12_cousub.zip; terms of use: https://www2.census.gov/geo/pdfs/maps-data/data/tiger/tgrshp2012/TGRSHP2012_TechDoc_Ch1.pdf).
(PNG)

**S7 Fig. Correlation of percentage of households in county subdivisions of Florida *not* connected to septic tanks with demographic and economic factors.** Data are from the Florida Department of Health septic tank inspection permits reported in 2012. Each point represents a county subdivision. Only county subdivisions with at least 5 households and 20 population size were included.
(PNG)

**S8 Fig. Correlation of percentage of households in county subdivisions of Florida in metropolitan statistical areas not connected to septic tanks with demographic and economic factors.** Data are from the Florida Department of Health septic tank inspection permits reported in 2012. Each point represents a county subdivision. Only metropolitan county subdivisions with at least 5 households and 20 population size were included.
(PNG)

**S9 Fig. Correlation of percentage of households in county subdivisions of Florida in micropolitan statistical areas not connected to septic tanks with demographic and economic factors.** Each point represents a county subdivision. Data are from the Florida Department of Health septic tank inspection permits reported in 2012. Only micropolitan county subdivisions with at least 5 households and 20 population size were included.
(PNG)

**S10 Fig. Correlation of percentage of households in county subdivisions of Florida in rural areas not connected to septic tanks with demographic and economic factors.** Data are from the Florida Department of Health septic tank inspection permits reported in 2012. Each point represents a county subdivision. Only rural county subdivisions with at least 5 households and 20 population size were included.
(PNG)

**S11 Fig. Fraction of present resident population receiving collection by county in 8 states.** Data are from the 2012 EPA Clean Watersheds Needs Survey. (a) Fraction shown on a continuous scale. (b) Fraction above and below 0.2. Only states that had more comprehensive responses in the survey are shown. Counties that did not report any data in the survey are shown in gray. Counties than report more than 100% of the present resident population receiving collection are shown in (a) as 1 and in (b) as $\geq$0.2. The map base layer is taken from the U.S. Census 2012 TIGER/Line Shapefile by U.S. County (https://www2.census.gov/geo/tiger/TIGER2012/COUNTY/tl_2012_us_county.zip; terms of use: https://www2.census.gov/geo/pdfs/maps-data/data/tiger/tgrshp2012/TGRSHP2012_TechDoc_Ch1.pdf).
(PNG)

**S12 Fig. Sensitivity analysis of Fig 5c from main text.** The effect on the time between population B peak and wastewater peak of varying the (a) population size of the two populations ($N_A$, $N_B$), (b) recovery rate ($\gamma_I$ in units of inverse days), and (c) basic reproduction number of the two populations ($R_0^A$, $R_0^B$) across a range of values. Unless where indicated, the parameters were $N_A = N_B = 5000$, $\gamma_I = 0.18$ days$^{-1}$, and $R_0^A$, $R_0^B = 1.5$.
(PNG)

**S13 Fig. Correlation between householder age and household income in the American Housing Survey dataset by census division.**
(PNG)

**S14 Fig. Sewer connectivity in Minnesota.** (a) Map showing whether a community in Minnesota (mapped to Census Designated Place) reported having a collection system in the 2021 Minnesota Wastewater Infrastructure Needs Survey. Almost all communities mapped to Census Designated Places. The map base layer is taken from the U.S. Census 2021 TIGER/Line Shapefile by Minnesota Census Designated Place (https://www2.census.gov/geo/tiger/ TIGER2021/PLACE/tl_2021_27_place.zip; terms of use: https://www2.census.gov/geo/pdfs/ maps-data/data/tiger/tgrshp2021/TGRSHP2021_TechDoc_Ch1.pdf). (b) Percentage of households without subsurface sewage treatment systems (i.e. without septic tanks, suggesting sewered) by county reported in the 2017 Subsurface Sewage Treatment Systems in Minnesota Annual Report [56]. Counties reporting more than 100% of households having subsurface sewage treatment systems are set at 100% (shown as 0% without subsurface sewage treatment systems in the map). The map base layer is taken from the U.S. Census 2017 TIGER/Line Shapefile by U.S. County (https://www2.census.gov/geo/tiger/TIGER2017/COUNTY/tl_2017_us_ county.zip; terms of use: https://www2.census.gov/geo/pdfs/maps-data/data/tiger/tgrshp2017/ TGRSHP2017_TechDoc_Ch1.pdf).
(PNG)

**S15 Fig. Comparison of the 2012 EPA Clean Watersheds Needs Survey (CWNS) and the 2019 and 2021 U.S.** Census American Housing Survey (AHS) datasets in core based statistical areas (CBSA). Data from the EPA CWNS are of the fraction of residents in the CBSA receiving sewage collection. Data from the AHS are of the fraction of households in the CBSA on sewer. Only CBSAs that were oversampled in the AHS data and were in states that had more comprehensive responses in the EPA CWNS were included (see Methods). The dashed black line shows y = x as a reference.
(PNG)

**S16 Fig. The percentage of the population that received wastewater collection by a municipal utility in Utah by county subdivision of Utah.** Data are from the 2021 Utah Municipal Wastewater Planning Survey. All values less than 0 are displayed as 0. County subdivisions with fewer than 5 households or 20 population size are displayed in white. Note that Indian reservations were not surveyed in this dataset. The map base layer is taken from the U.S. Census 2021 TIGER/Line Shapefile by Utah County Subdivision (https://www2.census.gov/geo/ tiger/TIGER2021/COUSUB/tl_2021_49_cousub.zip; terms of use: https://www2.census.gov/ geo/pdfs/maps-data/data/tiger/tgrshp2021/TGRSHP2021_TechDoc_Ch1.pdf).
(PNG)

**S17 Fig. Correlation of fraction households in county subdivisions of Utah with wastewater collection with demographic and economic factors.** Data are from the 2021 Utah Municipal Wastewater Planning Survey. Note that Indian reservations were not surveyed in this dataset. Values of the fraction of the county subdivision receiving collection greater than 1 were set to 1.
(PNG)

**S18 Fig. Fraction of households by Census Designated Place that are connected to sewers, septic tanks, or other forms of sewage disposal in the U.S. Island Areas.** (a) American Samoa, (b) Guam, (c) the Northern Mariana Islands, and (d) the Virgin Islands. Data are from

the 2020 U.S. Census Island Areas Decennial Survey.
(PNG)

**S19 Fig. Map of fraction of households connected to sewers by Census Designated Place.**
(a) American Samoa, (b) Guam, (c) the Northern Mariana Islands, and (d) the Virgin Islands.
Data are from the 2020 U.S. Census Island Areas Decennial Survey. The map base layers are
taken from the U.S. Census 2020 TIGER/Line Shapefile by American Samoa Census Desig-
nated Place (https://www2.census.gov/geo/tiger/TIGER2020/PLACE/tl_2020_60_place.zip),
Guam Census Designated Place (https://www2.census.gov/geo/tiger/TIGER2020/PLACE/tl_
2020_66_place.zip), Northern Mariana Islands Census Designated Place (https://www2.
census.gov/geo/tiger/TIGER2020/PLACE/tl_2020_69_place.zip), and Virgin Islands Census
Designated Place (https://www2.census.gov/geo/tiger/TIGER2020/PLACE/tl_2020_78_place.
zip). The terms of use for all maps can be found at: https://www2.census.gov/geo/pdfs/maps-
data/data/tiger/tgrshp2020/TGRSHP2020_TechDoc_Ch1.pdf.
(PNG)

**S1 Table. Additional datasets discussed in the supplementary information but excluded
from the main text due to data incompleteness and potential biases.**
(DOCX)

**S2 Table. Counties in the US with > = 5% of occupied housing units lacking complete
plumbing facilities.** Data are from the 2021 U.S. Census American Community Survey.
(DOCX)

**S3 Table. Correlation of the percentage of households in a county connected to public sew-
ers with different demographic or economic factors by state; same as Table 3, but includ-
ing all state-variable pairs.** q-values above 1 are set to 1.
(DOCX)

**S4 Table. Correlation of the fraction of households by county connected to public sewers
and different demographic or economic factors, stratified by state and Metropolitan,
Micropolitan, and Rural Statistical Areas.** Only correlations that are significant (q-value< =
0.05) are shown.
(DOCX)

**S5 Table. Manual changes community names reported in the Minnesota Wastewater Infra-
structure Needs Survey to be able to match with a U.S. Census Designated Place.**
(DOCX)

## Acknowledgments

We thank Cara Omana from the Minnesota Pollution Control Agency for generously sharing
the Minnesota WINS data and Harry Campbell from the Utah Department of Environment
Quality, Division of Water Quality for generously sharing the Utah MWPP survey data and to
both for answering questions about the datasets. We thank colleagues in the Grad lab and the
Harvard T. H. Chan School of Public Health Center for Communicable Disease Dynamics for
helpful discussions, particularly Stephen Kissler.

## Author Contributions

**Conceptualization:** QinQin Yu, Scott W. Olesen, Claire Duvallet, Yonatan H. Grad.

**Data curation:** QinQin Yu.

**Formal analysis:** QinQin Yu, Yonatan H. Grad.

**Funding acquisition:** Yonatan H. Grad.

**Investigation:** QinQin Yu, Yonatan H. Grad.

**Methodology:** QinQin Yu, Scott W. Olesen, Claire Duvallet, Yonatan H. Grad.

**Resources:** Yonatan H. Grad.

**Software:** QinQin Yu.

**Supervision:** Yonatan H. Grad.

**Validation:** QinQin Yu, Scott W. Olesen, Claire Duvallet.

**Visualization:** QinQin Yu, Scott W. Olesen, Claire Duvallet, Yonatan H. Grad.

**Writing – original draft:** QinQin Yu, Yonatan H. Grad.

**Writing – review & editing:** QinQin Yu, Scott W. Olesen, Claire Duvallet, Yonatan H. Grad.

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
