## [Decision Letter · Decision Letter 0]

15 Nov 2023

PGPH-D-23-01001

Assessment of sewer connectivity in the United States and its implications for equity in wastewater-based epidemiology

Dear Dr. Grad,

Thank you for submitting your manuscript to PLOS Global Public Health. After careful consideration, we feel that it has merit but does not fully meet PLOS Global Public Health’s publication criteria as it currently stands. Therefore, we invite you to submit a revised version of the manuscript that addresses the points raised during the review process.

We look forward to receiving your revised manuscript.

Kind regards,

Rajiv Sarkar

Academic Editor

Journal Requirements:

1. We ask that a manuscript source file is provided at Revision. Please upload your manuscript file as a .doc, .docx, .rtf or .tex.

Additional Editor Comments (if provided):

(1) The reviewers have made some important suggestions. Please respond to ALL comments by the reviewers.

(2) As pointed by reviewer 1, the rationale for a deep-dive using the Florida Department of Health onsite sewage treatment and disposal systems inspection data is unclear. Please provide a clear justification of the utility of this analysis for this paper.

(3) Please use the most recent datasets, wherever possible. If not, please discuss the potential limitations of using data that are more than a decade old (for example, the data from EPA Clean Watersheds Needs Survey is from 2012).

(4) Please provide more details about the justification of the model selection (SIR models vs other compartmental/meta-population models) and the assumptions for base parameter values used in the simulation models for exploring the applicability of WBE data to neighboring un-sampled communities as the findings have wider implications for wastewater-based surveillance.

Reviewers' comments:

Reviewer's Responses to Questions

**Comments to the Author**

1. Does this manuscript meet PLOS Global Public Health’s publication criteria? Is the manuscript technically sound, and do the data support the conclusions? The manuscript must describe methodologically and ethically rigorous research with conclusions that are appropriately drawn based on the data presented.

Reviewer #1: No

Reviewer #2: Yes

Reviewer #3: Yes

2. Has the statistical analysis been performed appropriately and rigorously?

Reviewer #1: No

Reviewer #2: Yes

Reviewer #3: Yes

3. Have the authors made all data underlying the findings in their manuscript fully available (please refer to the Data Availability Statement at the start of the manuscript PDF file)?

Reviewer #1: Yes

Reviewer #2: Yes

Reviewer #3: Yes

4. Is the manuscript presented in an intelligible fashion and written in standard English?

Reviewer #1: Yes

Reviewer #2: Yes

Reviewer #3: Yes

5. Review Comments to the Author

Reviewer #1: This paper provides an overview of the potential for WBE based on sewered connections in the United States. However, non-sewered sanitation systems can, and are also sampled and excluded from this analysis. Also missing is a comparison to how many sites are currently sampling for WBE compared to this research on the potential application.

1. Introduction is missing several key references and literature on this subject. At a minimum, suggest considered 3 additional references:

a. Burden of disease attributable to unsafe drinking water, sanitation, and hygiene in domestic settings: a global analysis for selected adverse health outcomes

b. The effects of racism, social exclusion, and discrimination on achieving universal safe water and sanitation in high-income countries

c. Using wastewater to overcome health disparities among rural residents

2. Methods, why was a convenience sample done and not the entire US as the title states? What were the criteria for this convenience. Why was Florida chosen as the dataset up to 2012 which is more than 10 years old.

3. I am surprised by the rural and urban dataset that is so old. WHO and UNICEF have new data out on the United States:

a. World Health Organization (WHO) and the United Nations Children’s Fund (UNICEF). Progress on household drinking water, sanitation and hygiene 2000-2022. 2023. https://data.unicef.org/resources/jmp-report-2023/

4. Section “Is there inequity in who is sampled by wastewater?” is not accurate to the data presented. This section is instead the portion of the population that could be sampled by WBE.

5. Have you considered also the role of CSO and SSO in Figure 1? This is important for dilution of samples.

6. The portion on sewer connectivity by racial group is missing important context for infrastructure inequalities. Without that, this section is not in context.

7. The section on Florida should be removed or add additional states. Otherwise, what is the context or rationale for a single state deep dive?

8. Table 2 Significance, round to 2 significant figures or state <.05. E-07 is too many significant figures.

9. Table 3 Significance, round to 2 significant figures or state <.05. Table 3 would be better to have all states.

10. In the section “Factors influencing applicability of wastewater-based epidemiology to communities lacking sewer connection” a lot more citations are needed to ground these assumptions. Reproduction number of 1.5 in particular. What other SIR models have been conducted (there have been many the authors are missing).

11. Compare the results of this work to how many locations are currently being sampled by BioBot or CDC NWSS for WBE.

Reviewer #2: The manuscript by Yu et al. addresses a timely and significant question regarding the spatial coverage of municipal based wastewater surveillance for infectious disease (although clearly their data and analysis could be applied for detection of non-microbial analytes, e.g. opioids, in wastewater). Although it is well-documented that there is a significant percentage of the U.S. population that is not connected to municipal wastewater, the “who” and “where” of this population is not well understood. The authors use a diverse set of data sources to better understand this population. This is important as sewersheds do not align with political or demographic boundaries and are determined by a diversity of factors, including topography.

Their descriptive data reproduces prior studies indicating that the assumption that the unsewered population would be uniformly more or less wealthy is incorrect—there is substantial diversity based on income, region, and locality, which reflects, at least in part, ethnicity. The use of multiple data sets and appropriate statistical analysis is an important advance on current knowledge. The data clearly show that first nation’s populations living in Alaska and the Navajo Nation are significantly overrepresented in being unconnected to municipal systems.

The modeling aspect of the study addresses a critical question in optimal coverage of a national surveillance system—“is the unsewered and thus unmonitored population informed by proximity and contact with a sewered and monitored population”? Although there is certainly room to expand this model to different levels of contact, including “mass participation events” such as sporting events or seasonal shifts in contact, the presented model is a valuable addition and lays the groundwork for future modeling and comparison to epidemiological data.

Reviewer #3: Overall, I believe this to be a broad analysis of factors contributing to potential inequities in the implementation of wastewater based epidemiology, and presents a summary of potential biases which would be reflected in a wastewater signal. My only major comment is that median income should be analyzed as a factor, particularly when some of the factors analyzed are likely to be nearly perfectly collinear.

Line 65-68: Wealthier with respect to local rural or urban area or wealthier compared to national statistics? Wording isn't great.

Line 125-127: Does sub-sampling for areas with a minimum number of households and population not bias your average sewered % up by omitting some potentially very rural areas? I would appreciate a % of total data included in the study.

Methods general: The number of observations used for your results should be included somewhere.

Results general: Some of the results suffer from the confounding effect of wealth. For example, householder age and the proposed correlation are likely to reflect average income and poverty, with age simply being colinear. Household income is part of the AHS, and should have been analyzed rather than the binary in poverty/not in poverty.

6. PLOS authors have the option to publish the peer review history of their article (what does this mean?). If published, this will include your full peer review and any attached files.

**Do you want your identity to be public for this peer review?** For information about this choice, including consent withdrawal, please see our Privacy Policy.

Reviewer #1: No

Reviewer #2: **Yes: **Guy Hughes Palmer

Reviewer #3: No

---

## [Decision Letter · Decision Letter 1]

29 Feb 2024

Assessment of sewer connectivity in the United States and its implications for equity in wastewater-based epidemiology

PGPH-D-23-01001R1

Dear Dr Grad,

We are pleased to inform you that your manuscript 'Assessment of sewer connectivity in the United States and its implications for equity in wastewater-based epidemiology' has been provisionally accepted for publication in PLOS Global Public Health.

Best regards,

Rajiv Sarkar

Academic Editor

Reviewer Comments (if any, and for reference):

Reviewer's Responses to Questions

**Comments to the Author**

1. If the authors have adequately addressed your comments raised in a previous round of review and you feel that this manuscript is now acceptable for publication, you may indicate that here to bypass the “Comments to the Author” section, enter your conflict of interest statement in the “Confidential to Editor” section, and submit your "Accept" recommendation.

Reviewer #1: All comments have been addressed

Reviewer #2: All comments have been addressed

Reviewer #3: All comments have been addressed

2. Does this manuscript meet PLOS Global Public Health’s publication criteria? Is the manuscript technically sound, and do the data support the conclusions? The manuscript must describe methodologically and ethically rigorous research with conclusions that are appropriately drawn based on the data presented.

Reviewer #1: (No Response)

Reviewer #2: Yes

Reviewer #3: Yes

3. Has the statistical analysis been performed appropriately and rigorously?

Reviewer #1: (No Response)

Reviewer #2: Yes

Reviewer #3: Yes

4. Have the authors made all data underlying the findings in their manuscript fully available (please refer to the Data Availability Statement at the start of the manuscript PDF file)?

Reviewer #1: (No Response)

Reviewer #2: Yes

Reviewer #3: Yes

5. Is the manuscript presented in an intelligible fashion and written in standard English?

Reviewer #1: (No Response)

Reviewer #2: Yes

Reviewer #3: Yes

6. Review Comments to the Author

Reviewer #1: (No Response)

Reviewer #2: The previous concerns raised by the reviewers have been satisfactorily addressed.

Reviewer #3: (No Response)

7. PLOS authors have the option to publish the peer review history of their article (what does this mean?). If published, this will include your full peer review and any attached files.

**Do you want your identity to be public for this peer review?** For information about this choice, including consent withdrawal, please see our Privacy Policy.

Reviewer #1: No

Reviewer #2: **Yes: **Guy Hughes Palmer

Reviewer #3: No
